# Codes between Poles: Linking Transcriptomic Insights into the Neurobiology of Bipolar Disorder

**DOI:** 10.3390/biology13100787

**Published:** 2024-09-30

**Authors:** Jon Patrick T. Garcia, Lemmuel L. Tayo

**Affiliations:** 1School of Chemical, Biological, and Materials Engineering and Sciences, Mapúa University, Manila 1002, Philippines; jptgarcia@mymail.mapua.edu.ph; 2School of Graduate Studies, Mapúa University, Manila 1002, Philippines; 3Department of Biology, School of Health Sciences, Mapúa University, Makati 1200, Philippines

**Keywords:** bipolar disorder, RNA-seq data, neurotransmitters, neurotransmission, disease-associated variants

## Abstract

**Simple Summary:**

Bipolar disorder is a psychiatric condition in which one prominent symptom is the regular occurrence of mood swings. Its root cause remains unclear; thus, this study was intended to provide new insights to better understand the genetics underlying such a disorder. Using samples obtained from three regions in the brain, it was found that some molecular and cellular mechanisms are involved in the disruption of neurobiological processes. The dysregulated expression of certain genes eventually leads to neurogenesis and neurotransmission impairment events in humans. Furthermore, these genes significant in the onset of bipolar disorder were identified and evaluated for the presence of variants, which may be targeted to engineer better curative treatment strategies for the disorder.

**Abstract:**

Bipolar disorder (BPD) is a serious psychiatric condition that is characterized by the frequent shifting of mood patterns, ranging from manic to depressive episodes. Although there are already treatment strategies that aim at regulating the manifestations of this disorder, its etiology remains unclear and continues to be a question of interest within the scientific community. The development of RNA sequencing techniques has provided newer and better approaches to studying disorders at the transcriptomic level. Hence, using RNA-seq data, we employed intramodular connectivity analysis and network pharmacology assessment of disease-associated variants to elucidate the biological pathways underlying the complex nature of BPD. This study was intended to characterize the expression profiles obtained from three regions in the brain, which are the nucleus accumbens (nAcc), the anterior cingulate cortex (AnCg), and the dorsolateral prefrontal cortex (DLPFC), provide insights into the specific roles of these regions in the onset of the disorder, and present potential targets for drug design and development. The nAcc was found to be highly associated with genes responsible for the deregulated transcription of neurotransmitters, while the DLPFC was greatly correlated with genes involved in the impairment of components crucial in neurotransmission. The AnCg did show association with some of the expressions, but the relationship was not as strong as the other two regions. Furthermore, disease-associated variants or single nucleotide polymorphisms (SNPs) were identified among the significant genes in BPD, which suggests the genetic interrelatedness of such a disorder and other mental illnesses. *DRD2*, *GFRA2*, and *DCBLD1* were the genes with disease-associated variants expressed in the nAcc; *ST8SIA2* and *ADAMTS16* were the genes with disease-associated variants expressed in the AnCg; and *FOXO3*, *ITGA9*, *CUBN*, *PLCB4*, and *RORB* were the genes with disease-associated variants expressed in the DLPFC. Aside from unraveling the molecular and cellular mechanisms behind the expression of BPD, this investigation was envisioned to propose a new research pipeline in studying the transcriptome of psychiatric disorders to support and improve existing studies.

## 1. Introduction

Manic depression, commonly known as bipolar disorder (BPD), is a serious mental health condition characterized by a recurrent shifting of mood between two emotional polarities, which are mania or hypomania (highs) and depression (lows) [1]. Globally, around 2.4% of the population is clinically diagnosed with BPD [2]. No cure has been discovered for the disorder yet, although current medications have shown promising results for regulating its symptoms. Several studies have already been performed that elucidate the genetic underpinnings of BPD, but there are only limited studies discussing the manifestations of the disorder at the transcriptomic level. Krebs et al. performed differential expression analysis (DEA) of the transcriptomic profiles obtained from whole blood samples in which the results suggested the possible role of lithium in the cellular mechanisms pertinent among BPD patients [3]. Moreover, it was observed from a study by Holmgren et al. that dysregulation of stoichiometric balance has implications for the pathology of BPD [4]. On the other hand, Ardesch et al. analyzed the normative expression profiles of risk genes linked to four neuropsychiatric disorders, including BPD, and concluded brain dysfunction as one of the shared pathways among the illnesses [5]. Most transcriptomic studies of this disorder are commonly investigated as a group, along with other brain conditions [6,7,8]. However, when studied in isolation, some of these only present the genes and pathways underlying the disorder but fail to interpret the systematic relationship of gene sets to elucidate possible molecular and cellular associations in bringing new insights and routes for inventing novel medications or improving existing treatments against the deregulated expression of BPD.

The past decades have shown a growing interest in the use of RNA sequencing as a technique in the study of transcriptomics. As a newer and more advanced technology compared to conventional sequencing methods such as microarrays, RNA-seq data utilizes next-generation sequencing (NGS), which provides more accurate and reliable results that could lead to deeper insight into the transcriptome [9,10]. Most transcriptomic studies of BPD used blood samples for sequencing; however, this raises concerns as whole blood may confer limitations in explaining the deviations in brain function among people with the disorder [11,12,13]. However, it is undeniable that blood is commonly utilized for sampling because it is more accessible and less invasive than other biological sources. To provide a more holistic understanding of the manifestations of BPD, post-mortem examination of brain samples may be favored. A study by Luykx et al. performed RNA sequencing and DEA of medial frontal gyrus brain tissues to determine the implication of protein-coding and lncRNAs among patients with BPD [14]. Furthermore, using hippocampal and orbitofrontal cortex samples, Darby et al. highlighted the potential of mRNAs and protein synthesis as therapeutic targets for treating BPD, schizophrenia (SCZ), and major depressive disorder (MDD) [15]. Ellis et al., on the other hand, utilized cortical brain tissues of patients with autism, SCZ, and BPD to quantify the extent of genetic correlation among these three [16]. Similar to what was mentioned above, most of these studies analyzed and compared BPD relative to other psychiatric disorders. Therefore, results from the current literature may not support medical scenarios in which BPD is regarded as an independent case.

Unfortunately, the use of post-mortem brain samples possesses considerable limitations since the rate of degradation of the tissue can incur a significant impact on the analysis. This method is greatly time-dependent; therefore, the post-mortem interval (PMI), which is the interval between the time of death and the time the sample will be analyzed, must be kept short. In the study of Liharska et al., they inferred that post-mortem brain samples should not be used as a proxy for living brain samples because both show vast differences in expression profiles [17]. However, such a conclusion was considered premature and unsupported by relevant data. True enough that RNA transcripts readily start degrading shortly after death, but this is not an absolute reason to completely consider post-mortem brain samples as insignificant for human brain research. In fact, several studies suggest otherwise. Fromer et al. found that expression profiles sequenced from post-mortem brain samples have a genetic correlation with the occurrence of brain disorders [18]. Furthermore, transcriptomic analyses performed in the studies of Raj et al. and Gandal et al. showed that predicted associations of genetic underpinnings among certain diseases were proved factual based on the differential expression of genes in post-mortem-controlled cases [19,20].

With the general objective of providing a deeper understanding of the molecular and cellular causes of BPD, this study aimed to perform intramodular connectivity analysis and network pharmacology assessment of diseases-associated variants using different samples from three regions in the brain: the nucleus accumbens (nAcc), the anterior cingulate cortex (AnCg), and the dorsolateral prefrontal cortex (DLPFC). We intended to use post-mortem samples that have been previously RNA-sequenced to contribute more relevant information into the transcriptome of BPD and further support existing studies with a similar goal but utilizing whole blood transcripts. The intramodular connectivity analysis was employed to elucidate what gene expression patterns are highly correlated to which region of the brain that would give insights into their specific roles and present potential therapeutic targets for treating the disorder. Aside from that, network pharmacology was carried out to determine which genes from these regions are greatly associated with BPD. These were then further investigated by obtaining their first-degree neighbors and identifying the presence of disease-associated variants or single nucleotide polymorphisms (SNPs) that would explain the genetic interrelatedness of BPD and other psychiatric disorders.

## 2. Materials and Methods

### 2.1. Nature of the Dataset

The RNA-seq data utilized in this study were obtained from the National Center for Biotechnology Information–Gene Expression Omnibus (NCBI–GEO) [https://www.ncbi.nlm.nih.gov/geo/ (accessed on 15 June 2024)]. GEO from NCBI is an online repository of genomics data from various research and medical institutes where sequencing data of a vast range of organisms are stored for public use. GSE80655 was the chosen dataset derived from a previous study by Ramaker et al. [21]. Only data pertaining to BPD were included, and the rest were excluded. Twenty-four post-mortem brain tissues of human (*Homo sapiens*) patients diagnosed with BPD, with each sample obtained from the nAcc, AnCg, and DLPFC regions, were sequenced using Illumina HiSeq 2000, equating to a total of 72 samples used for this study. A statistical description of the samples to explain in detail the clinical data controls and demographic profiles of their sources is presented in Appendix A.

### 2.2. Pre-Processing of Dataset

The raw dataset was loaded into the R program v4.2.3 [https://cran.r-project.org/bin/windows/base/ (downloaded on 21 June 2024)], where the initial computation of the dataset was performed. Using Bioconductor v3.19 [https://www.bioconductor.org/ (accessed on 21 June 2024)], which is a free, open-source software for facilitating comprehensive analyses of biological assays, we filtered and modified the dataset for quality control before it was subjected to intramodular analysis. The “goodSamplesGenes” function was employed to check and remove genes with zero variance and missing values or entries. Moreover, hierarchal clustering using the “hclust” function was performed to detect any outlying samples. Principal component analysis was then followed to provide a more quantitative basis for excluding the outliers. After which, DESeq2 was utilized to examine the expression patterns among the genes from the RNA-seq data, and all those genes with counts of less than 15 in more than 75% of the samples were removed.

### 2.3. Intramodular Connectivity Analysis

Variance stabilization and normalization were performed as the final steps before proceeding with WGCNA. Using the “pickSoftThreshold” function, the network topology was constructed to determine the soft-thresholding power (*β*) to which the co-expression similarity values of the genes will be raised. The mean connectivity and correlation coefficient values were extracted and plotted to visualize the power of choice based on a scale-free topology model fit limit of 80% to ensure that the adjacency matrices that were calculated in estimating the expression modules were highly accurate. Furthermore, the “blockwiseModules” function was employed using a block size of 20,000 to obtain the modules that were constructed from the correlation of genes with respect to the calculated module eigengenes. The “plotDendroAndColors” function was then used to plot the distribution of the modules in the dendrogram and describe how these were identified from the hierarchal clustering of the genes. After which, the respective module membership and *p*-value of each gene were calculated, and the categorical variables, the nAcc, AnCg, and DLPFC, were binarized to compute the module–gene associations that would elucidate what genetic expressions have high intramodular connectivity. Finally, using “CorLevelPlot”, a heatmap was generated to determine only the significant modules that would accurately characterize any of the three brain regions.

### 2.4. Functional Annotation and Pathway Enrichment of Modules

From the heatmap, those modules labeled in red with three asterisks were recorded and subjected to functional annotation and pathway enrichment. The higher the *p*-value, the darker the color of the module, which suggests a greater positive correlation among the genes grouped in that module. Meanwhile, the higher the number of asterisks, the more likely that that region will characterize the expression patterns in that module. Each module was then imported to Cytoscape v.3.10.2 [https://cytoscape.org/index.html (downloaded on 15 July 2024)] to construct their PPI network under a confidence level of 0.90 with the aid of stringApp v2.1.1 [https://apps.cytoscape.org/apps/stringapp (downloaded on 10 July 2024)]. From the generated large network of interactions, the top 50 hub genes were determined using the degree topological algorithm from cytoHubba v0.1 [https://apps.cytoscape.org/apps/cytohubba (downloaded on 15 July 2024)]. Subsequently, the hub genes of each module were sent to ShinyGO v0.80 [http://bioinformatics.sdstate.edu/go/ (accessed on 17 July 2024)] for functional annotation and pathway enrichment [22]. These gene sets were characterized based on three gene ontology classifications: biological process (BP), cellular component (CC), and molecular function (MF). Aside from that, KEGG pathway analysis was employed to further identify the expression patterns in the modules [23,24]. An FDR cutoff of 0.05 was used, and any redundant category among the lists was removed. Only those clusters with the highest fold enrichment score from each annotation were sent to the SRplot web server [http://www.bioinformatics.com.cn/srplot (accessed on 16 July 2024)] for enrichment visualization.

### 2.5. Network Pharmacology Assessment of Disease-Associated Variants

The gene significance of the expression profiles obtained from the three brain regions was calculated and ranked amongst each other. The top 100 genes from the nAcc, AnCg, and DLPFC were recorded and sent to DisGeNET [https://disgenet.com (accessed on 17 July 2024)] for disease-association analysis [25]. Consequently, those genes that are found positive for disease-associated variants or SNPs were further evaluated by determining their first-degree neighbors using a whole-genome PPI network constructed from the expression profiles of BPD patients under an interaction score of 0.7.

## 3. Results

### 3.1. Intramodular Connectivity Analysis

After pre-processing the raw RNA-seq data, the network topology and mean connectivity were calculated to determine the soft-thresholding power that was used to identify the modules. Appendix A shows the scale-free topology model fit and mean connectivity plots of the dataset from which the soft-thresholding power was chosen. Since *β* = 12 obtained the highest mean connectivity among the values that reached the 80% threshold, it was utilized to calculate the gene co-expression networks. From there, the module eigengenes were analyzed, and each gene was quantitatively classified based on their connectivity relative to the module eigengene to construct the modules that significantly characterize the expression profiles prominent among the three brain regions. Figure 1 displays the dendrogram, which describes how the modules were identified based on the height of the branches that correlate to the strength of connectivity among sets of genes.

A total of 25 modules, including the grey module, were obtained from the WGCNA. Furthermore, the selection was filtered down by calculating the respective module membership and *p*-value of the genes within each module. Seven out of the twenty-five modules showed high intramodular connectivity. The pink, dark green, grey60, red, and purple modules were highly correlated to the nAcc, while the turquoise and blue modules were highly correlated to the DLPFC. The AnCg did not show a high correlation to any of the modules, as observed by the absence of modules with three asterisks. Figure 2 presents the module–gene connectivity heatmap of the samples wherein positive correlations are labeled in pink while negative correlations are labeled in blue. The darker the color of the module, the greater the connectivity among its genes. Moreover, the number of asterisks indicates the likelihood that the module is highly expressed in that region. A greater number of asterisks in that module suggests greater intramodular connectivity, or in other words, a higher probability of that certain expression occurring in that region.

### 3.2. Functional Annotation and Pathway Enrichment

The seven modules obtained from the previous analysis were sent to Cytoscape to obtain their top 50 hub genes using the degree algorithm for functional annotation and pathway enrichment. The hub genes were then sent to ShinyGO to characterize the modules based on BP, CC, and MF annotations. These were also subjected to KEGG pathway analysis to determine the possible routes to which these sets of genes are similar. Figure 3 presents the annotation bar plots of the cluster with the highest fold enrichment from each module. The top five highest clusters are also listed in Appendix A with their respective enrichment FDR score, ID, pathway description, and genes.

In the BP annotation, the blue module corresponded to *synaptic vesicle exocytosis*; the dark green module to *epithelial cilium movement involved in extracellular fluid movement*; the grey60 module to *myeloid leukocyte activation*; the pink module to *epinephrine metabolic process*; the purple module to *Golgi lumen acidification*; the red module to *carboxylic acid catabolic process*; and the turquoise module to *mitochondrial electron transport ubiquinol to cytochrome c*. In the CC annotation, the blue module corresponded to *synaptobrevin 2-SNAP-25-syntaxin-1a-complexin I complex*; the dark green module to *outer dynein arm*; the grey60 to *tertiary granule membrane*; the pink module to *beta-catenin-TCF complex*; the purple module to *proteosome regulatory particle lid subcomplex*; the red module to *heterotrimetric G-protein complex*; and the turquoise module to *mitochondrial respiratory chain complex III*. In the MF annotation, the blue module corresponded to *phosphatidylinositol-3,4-bisphosphate 5-kinase activity*; the dark green module to *minus-end-directed microtubule motor activity*; the grey60 module to *ICAM-3 receptor activity*; the pink module to *opioid peptide activity*; the red module to *glutamate dehydrogenase (NAD+) activity*; and the *turquoise module to succinate dehydrogenase (ubiquinone) activity*.

On the other hand, Figure 4 shows the bubble plot of the cluster with the highest enrichment FDR from each module obtained from the KEGG pathway analysis.

The top three clusters associated with the blue module were *insulin secretion*, *synaptic vesicle cycle*, and *inflammatory mediator regulation of TRP channels*; the dark green module were *Huntington's disease*, *amyotrophic lateral sclerosis*, and *pathways of neurodegeneration-multiple diseases*; the grey60 module were *staphylococcus aureus infection*, *complement and coagulation cascades*, and *leishmaniasis*; the pink module were *tyrosine metabolism*, *cocaine addiction*, and *serotonergic synapse*; the purple module were *vibrio cholerae infection*, *homologous recombination*, and *proteasome*; the red module were *histidine metabolism*, *beta-alanine metabolism*, and *nitrogen metabolism*; and the turquoise module were *longevity regulating pathway-multiple species*, *EGFR tyrosine kinase inhibitor resistance*, and *ErbB signaling pathway*.

### 3.3. Network Pharmacology Assessment of Disease-Associated Variants

The genes with the highest significance value among the nAcc, AnCg, and DLPFC were sent to DisGeNET for disease-association analysis. In the nAcc, the genes found to be associated with BPD were *DRD2*, *PPP1R1B*, *CHRNA2*, *GFRA2*, *DCBLD1*, *PENK*, *NOS1AP*, *PART1*, and *NRN1*; in the AnCg, the genes found to be associated with BPD were *GRIK4*, *ST8SIA2*, *FABP7*, *TSHZ1*, *ADRA1A*, *ADAMTS16*, *HTR7*, *PCDH17*, and *PNPLA3*; and in the DLPFC, the genes found to be associated with BPD were *IGFBP2*, *SLC17A6*, *FOXO3*, *ITGA9*, *CARTPT*, *PREP*, *NEFM*, *CUBN*, *PLCB4*, *NOS1*, *NOS2*, *FGF9*, *RORB*, *SGCG*, and *ZNF365*. Moreover, these genes were further evaluated for the presence of SNPs. Among the reported genes in the nAcc, *DRD2*, *GFRA2*, and *DCBLD1* presented potential SNP interference; among the reported genes in the AnCg, *ST8SIA2* and *ADAMTS16* presented potential SNP interference; and among the reported genes in the DLPFC, *FOXO3*, *ITGA9*, *CUBN*, *PLCB4*, and *RORB* presented potential SNP interference. Table 1 shows the summary of disease-associated variants found among the three brain regions. Meanwhile, the complete list of significant genes, along with their associated SNPs, is listed in Appendix A.

Furthermore, those genes with disease-associated variants were sent again to Cytoscape to determine their first-degree neighbors using the PPI network constructed from the expression profiles of BPD patients. Figure 5 displays the network interactions of the first-degree neighbors of *DRD2*, which was the gene with the highest GDA score among the genes with SNPs found in the three brain regions. The PPI network of the other genes is shown in Appendix A.

## 4. Discussion

### 4.1. Genetic Evidence of Bipolar Disorder Occurrence

Contrary to popular belief, genetics does play a major role in the increased risk of BPD in children who happen to have parents with the disorder. Although environmental stimulation indeed aggravates the manifestation of BPD, disparities in the transcriptome among positive patients relative to healthy individuals present strong evidence of its genetic cause. Recent discoveries from genomic studies of BPD have further proven its intricate and complex nature. A genome-wide association study (GWAS) by Stahl et al. found 30 loci that were genome-wide significant, of which 20 were newly reported [26]. Moreover, a much earlier GWAS conducted by the Psychiatric Genomics Consortium, involving 41,917 BPD patients and 371,549 negative controls, revealed 64 genomic locations that are associated with an increased risk of BPD, 33 of which were never found in previous literature [27]. These results greatly improved the initial findings of the former study, bringing better insights into determining more risk locations in the DNA. Furthermore, a study by the Stanley Center for Psychiatric Research compared the exomes of 14,000 BPD patients against 14,000 negative controls with the hope of identifying the sequences that cause dysregulation in the synthesis of certain proteins in BPD [28]. The analysis showed that the *AKAP11* gene is a strong genetic factor that increases the risk of having the disorder.

These are only some of the several studies that prove the genetic underpinnings of BPD. Newer technologies can bring more novel findings that could further the scientific understanding of such disorders, especially since researchers now are not limited to using genomic approaches. Analyzing the transcriptome could also shed light on discovering molecular and cellular mechanisms that are evident in its occurrence. Moreover, delving into certain samples, such as the brain, where psychiatric disorders are most likely linked, may aid in identifying specific regions that would be best targeted for treating BPD.

### 4.2. Deregulated Transcription of Neurotransmitters in the Nucleus Accumbens

Adjacent to the septal nuclei, the nAcc is a region that is part of the ventral striatum, located in the rostrobasal forebrain [29,30]. It acts as the neural interface between motivation and action and is considered a significant part of the reward system of the brain [31]. Dysfunction of the nAcc has shown associations with various psychiatric disorders (i.e., obsessive–compulsive disorder, addiction, and MDD) [32,33,34]. The disruption of reward processing in mood disorders, such as that in MDD, can lead to anhedonia, which is the inability to perceive pleasure, and it is caused by an abnormal function of the nAcc [35]. Atypical stimulation of stressors in this region may also prime anxiety disorders (ADs), ultimately influencing emotional and behavioral responses. Moreover, the fluctuating activity of the nAcc in the brain can affect the instability of mood, a common manifestation in BPD, in which higher activation may exacerbate manic periods while lower activation may lead to depressive episodes [36]. Thus, in a variety of psychiatric diseases, dysregulation of the nAcc can take a toll on the intricate molecular and cellular interactions that drive motivation, emotional regulation, and reward processing.

It has long been known how important neurotransmitters are in neural homeostasis. Some of these neurotransmitters that play a significant function in the manifestation of mood swings in BPD are dopamine, serotonin, and norepinephrine. The manic polarity of BPD is associated with elevated levels of dopamine and increased norepinephrine activation, which result in impulsivity and the feeling of euphoria [37]. Meanwhile, the depressive polarity of BPD is linked to low levels of serotonin and increased secretion and conversion of norepinephrine, which result in irritability and the feeling of melancholy [38]. The complex interplay of these neurotransmitters, along with glutamate, an excitatory neurotransmitter, and gamma-aminobutyric acid (GABA), an inhibitory neurotransmitter, further underscores the multifaceted nature of the disorder [39,40]. Unlike other neurotransmitters, epinephrine is rarely mentioned in the neurobiology of BPD. Dopamine and norepinephrine, which are the precursors of epinephrine, are more involved in the central nervous system (CNS), which explains their direct effect on the brain. The blood–brain barrier (BBB) is impermeable against epinephrine, making it a more essential neurotransmitter in the peripheral nervous system (PNS) [41]. Dopamine and norepinephrine are significantly synthesized in the brain through the amino acid tyrosine, and their relative concentrations greatly affect the shifting of moods in BPD. Although epinephrine could be synthesized from dopamine and norepinephrine inside the brain, it is only in small amounts and its effects are less pronounced than its precursors. The genes found to be associated with the *epinephrine metabolic process* in the BP annotation simply suggest that the dysregulation of the nAcc could be influencing the elevated levels of dopamine and norepinephrine in the brain. This notion implies that the nAcc might be one of those overstimulated regions that contribute to the increased concentrations of these neurotransmitters, causing manic episodes in BPD. Otherwise, epinephrine metabolism would not occur if there was an absence of dopamine and norepinephrine.

The β-catenin/TCF complex is a transcriptional regulator that activates the signaling of Wnt target gene transcription in the nucleus [42]. This complex is essential in synthesizing the proteins necessary for the normal functioning of the body, thus, the disruption in its regulation may lead to serious disorders. Impairment in the Wnt pathway has been associated with various psychiatric disorders in humans, such as *WNT1*, *WNT2*, and *WNT7A* in autism spectrum disorder [43], *WNT7B* and *LRP5* in SCZ [44], and *LRP5* and *LRP6* in attention-deficit/hyperactivity disorder [45]. In BPD, *WNT2B* and *WNT7A* are reported to be involved [46]. The β-catenin/TCF complex has major implications on neurotransmission and neurogenesis in the brain. On the one hand, the interaction of β-catenin with TCF transcription factors influences the expression of genes involved in neurotransmission and synaptic function [47]. Deviations occurring in these interactions may result in deregulated secretion and activation of neurotransmitters that eventually manifest in BPD. On the other hand, signaling pathways induced by the β-catenin/TCF complex are important in promoting the differentiation of neural progenitor cells during neurogenesis [48]. Impairment of these signaling pathways could lead to structural brain changes; this is characterized by smaller hippocampal volumes reported in patients with BPD [49]. The genes found to be associated with the *beta-catenin-TCF complex* in the CC annotation simply suggest that problems in transcription regulation could be occurring in the nAcc. This further explains that the dysregulation of Wnt pathways in this brain region might be a cause of the unrestricted production and release of neurotransmitters in BPD, that is, signifying its relationship with the *epinephrine metabolic process* in the BP annotation.

Opioid peptides, including endorphins, enkephalins, and dynorphins, are also neurotransmitters that interact with various opioid receptors in the brain and nervous system to modulate a wide range of neural processes. In psychiatric disorders, opioid peptides are implicated in the deregulation of emotional and behavioral responses [50]. Just like the role of dopamine, serotonin, and norepinephrine in the occurrence of mood swings in BPD, concentration levels of opioids in the brain also characterize the complex shifting patterns between mania and depression [51]. Moreover, opioid peptide–receptor interactions regulate the activation of the hypothalamus–pituitary–adrenal (HPA) gland, and this influences the production and secretion of crucial neurotransmitters in BPD [52]. Although there are still limited studies that back up the role of the opioid system in the symptoms of BPD, a few pieces of literature have paved solid and promising evidence of its genetic foundation. In the study of Qi et al., they concluded that *OPRM1*, a Mu-opioid receptor (MOR) gene, is one of the candidate biomarkers for the increased risk of BPD [53]. Another study by Lindtröm et al. and Genazzani et al. reported that a fluctuation of endorphin levels in the cerebrospinal fluid (CSF) is manifested among manic patients [54,55]. The genes found to be associated with *opioid peptide activity* in the MF annotation simply suggest the important role of these neurotransmitters in the molecular etiology of BPD. Along with the result from the BP annotation, this greatly enlightens how the dysregulation of the transcription process in the nAcc affects the mechanistic interplay of nervous and endocrine ligand–receptor complexes in the expression of BPD.

In the KEGG pathway analysis, the pink module presented genes from the nAcc responsible for *tyrosine metabolism*. This implies that there may be disruption in the metabolic pathway of tyrosine in the body, which could further explain the deregulated conditions of neurotransmitter secretion manifested in BPD. Tyrosine is an essential element in the production of dopamine and norepinephrine. Interference in the metabolic activity of tyrosine can negatively impact the synthesis of dopamine, which could eventually affect norepinephrine concentrations in the brain [56,57].

### 4.3. Neurotransmission Impairment in the Dorsolateral Prefrontal Cortex

The DLPFC is a region located in the frontal lobe, in the prefrontal cortex specifically, that is responsible for the higher cognitive functions of the brain. It is adjacent to the limbic system, which is involved in the regulation of behavior and emotion [58]. The DLPFC often exhibits decreased activity during manic periods, which may impede the capacity of the brain to manage impulses. It is also associated with a reduction in inhibitory control that can exacerbate risk-taking behaviors and impair judgment [59]. Meanwhile, during depressive conditions, the DLPFC becomes inefficient in interacting with other brain regions responsible for mood regulation. This leads to hypoactivity, which, in turn, results in difficulties in concentration and decision-making [60]. Aside from modulating emotional responses, the DLPFC has become an interest in the study of BPD because of its implications in the cognitive aspect of the brain, and the dysregulation of expression patterns in this region may have significant effects on certain manifestations of the disorder.

Neurotransmitters, the chemical messengers involved in neurotransmission, are crucial for regulating various brain functions and processes. Imbalance or dysfunction in neurotransmitter systems can lead to the symptoms expressed among psychiatric disorders. Synaptic vesicle exocytosis is a fundamental element in the transmission process of nerve impulses to send the necessary signals to and from neurons to respond to external stimuli [61]. Signal neurotransmission occurs when neurotransmitters enclosed in a vesicle diffuse across the synaptic cleft and bind to receptors in the postsynaptic membrane to be delivered from one neuron to another. This interaction results in the opening of ion channels which generate electrical signals to carry out cellular responses. Dysregulation of such process could lead to neural miscommunication that may result in unrestricted activation, secretion, production, or differentiation of various neurotransmitters, in which increased or decreased concentrations of these chemicals could alter the normal functioning of the brain.

Moreover, impairment in synaptic exocytosis could also be a result of dysfunctional transmembrane proteins responsible for the specific attachment of exosomes. The SNARE complex is an important cellular component in exocytotic activities during neurotransmission, and alterations in its function may contribute to the neurochemical imbalances that underlie the characteristic mood swings and emotional instability in BPD. Three key proteins make up the SNARE complex: synaptobrevin, syntaxin, and SNAP-25. Synaptobrevin is found on the synaptic vesicle, while syntaxin and SNAP-25 are found on the presynaptic membrane [62]. In BPD, unregulated neurotransmission may cause an abnormal relay of signals between neurons, and this might be potentially correlated to an overactive or dysfunctional SNARE complex [63]. As a consequence, neurotransmitters become unregulated in the brain and manifest into various mood symptoms.

The genes found to be associated with *synaptic vesicle exocytosis* in the BP annotation, *synaptobrevin 2-SNAP-25-syntaxin-1a-complexin I complex* in the CC annotation, and *phosphatidylinositol-3,4-bisphosphate 5-kinase activity* in the MF annotation simply suggest that in BPD, the DLPFC could be characterized by the expression of genes responsible for the disruption of neurotransmission processes prominent in the brain. This notion may be a reason why, during either manic or depressive episodes, the DLPFC exhibits molecular and cellular disjunction with other brain regions that lead to poor cognitive functions. This could also be a cause of dysregulated phosphorylation of proteins, resulting in the eventual disruption of the said process. Such involvement of the DLPFC proves its role in aggravating the complex mood patterns observed in BPD.

Furthermore, the KEGG pathway analysis showed that the DLPFC may express genes associated with *insulin secretion*. The excitatory and inhibitory impulses in the brain are stabilized by insulin receptors on presynaptic neurons, which regulate neurotransmitter release. Insulin facilitates the translocation of glucose transporters to the cell surface, which is essential for glucose uptake in neurons and glial cells [64]. This ensures that neurons have enough energy for generating action potentials, recycling synaptic vesicles, and synthesizing neurotransmitters, especially glutamate and GABA, which greatly rely on glucose metabolism [65]. These genes determined from the DPLFC suggest the presence of irregularities in the secretion pathway of insulin in the body, and this problem may interfere with the generation of action potentials between neurons, which could be one component in the manifestation of neurotransmission impairment in BPD.

### 4.4. Risk Genes and Polymorphisms in Bipolar Disorder

Scientists have now been targeting gene candidates and biomarkers that present the risk of BPD at the earlier stages. From the analysis, among the genes with the highest significance value, we found that the nAcc has the greatest association with *DRD2*, the AnCg with *GRIK4*, and the DLPFC with *IGFBP2*.

*DRD2* (dopamine receptor D2) is an important component in the dopamine system of the body [66]. Alterations in this gene have associated it with various psychiatric disorders such as SCZ and drug addiction disorders. In a study by Huang et al., they reported *DRD2* as one of the candidate genes in the etiology of BPD [67]. *DRD2* interacts with *ANKK1* to initialize a certain dopaminergic pathway that causes the differentiation of BPD-1 and BPD-2. Similar results were observed in the study of Lee et al., wherein they concluded that disparities in the interaction between *DRD2/ANKK1* and *ALDH2* genes characterize the subtypes of BPD [68].

Moreover, *GRIK4* (glutamate receptor, ionotropic kainite 4) is a type of ion channel that facilitates excitatory neurotransmission processes in the brain by binding to glutamate, a neurotransmitter [69]. One of the prominent effects of a dysfunctional *GRIK4* is the disruption of glutamatergic signaling, which is essential for efficient neurotransmission and synaptic plasticity. Knight et al. reported that the reduced expression of *KA1*, as a result of the absence of a deletion variant in *GRIK4,* was associated with an increased risk of BPD [70]. *KA1* is essential in the increase of *GRIK4* mRNA abundance which mitigates symptoms of BPD. A 3′ UTR deletion variant that exhibited protection against this disorder was also identified by Pickard et al. [71]. The results of their study suggested that an increase in kainate receptor activity caused by drugs, which can mimic the expression-modifying effect of such a deletion variant, may offer a treatment target for BPD and other psychiatric disorders.

Meanwhile, *IGFBP2* (insulin-like growth factor binding protein 2) is found to be involved in certain cellular mechanisms in the brain that are vital in neurogenesis and neuroprotection. Alterations in this gene have shown associations with neuroinflammation and neurodegeneration, resulting in a loss of emotional and cognitive function among multifarious psychiatric disorders [72]. In the study conducted by Bezchlibnyk et al., they observed reduced expressions of *IGFBP2* in patients with BPD relative to controls, suggesting the role of this gene in the pathophysiology of the disorder [73]. They also reported a greater reduction of expression profiles in patients who were not treated with lithium, which further supports their claim regarding the function of *IGFBP2* in the etiology of BPD. Additionally, the same conclusions were drawn from the study of Fernandez and Torres-Alemán, in which they stated that the unregulated expression of *IGFBP2* has implications for the proper development of the brain, thereby affecting the survival, proliferation, and differentiation of neurons [74].

The presence of SNPs can impact how the manifestations of BPD are phenotypically expressed, such that these can affect the onset and intensity of its symptoms. These genetic variations may be linked to an increased risk of rapid emotional cycling or mood episodes, more severe manic or depressive periods, or earlier inception of the condition. Due to the vast range of clinical symptoms caused by SNPs, BPD diagnosis and treatment can become more challenging.

One study presented the association of BPD with multiple SNPs of the *IL1B* and *IL6R* genes [75]. The researchers proved that the impact of SNPs in BPD has a potential influence on the relationship between the immune and nervous systems, which can characterize molecular and cellular mechanisms involved in psychiatric disorders. The presence of SNPs can also explain why BPD, in any way, is correlated to other mood disorders such as SCZ and MDD. These genetic variations can modify biological pathways in the brain that may result in similar mechanisms found in other disorders. As previously mentioned, in the study of Mikhalitskaya et al., they found rs2228145, SNP of the *ILFR6* gene, to be linked with BPD, SCZ, and MDD [75]. de Marco et al., on the other hand, recorded 66 SNPs among 29 genes that have associations with both BPD and substance use disorder [76]. Some of these were rs11600996 of the *ARNTL* gene, rs6971 of the *TSPO* gene, and rs945032 of the *BDKRB2* gene. Moreover, from the genome assessment analysis conducted by Wang et al., they reported the presence of rs11789399, SNP of the *ASTN2* gene, in both BPD and SCZ cases [77].

SNPs contribute to the high rates of comorbidity and clinical overlap between BPD and other psychiatric conditions [78]. For instance, individuals with BPD may exhibit symptoms of anxiety aside from manic and depressive episodes. Such SNPs responsible for the disruption of genes that control mood, stress response, and emotional regulation could also increase the likelihood of developing AD or MDD. Thus, studying the transcriptome of BPD is one way of evaluating the genetic overlap among psychiatric disorders to assess biomarkers that could be targeted for therapy.

Most available drugs in the market are designed to target only specific proteins: for instance, remoxipride, which is an antipsychotic drug selective for dopamine D2 receptors [79], ketamine, which is a dissociative anesthetic that targets ionotropic glutamate receptors [80,81], and mecasermin, which is used as a treatment for growth failure in patients with IGF-1 deficiency [81]. These drugs are commonly utilized for maintaining proper regulation of disrupted proteins and not as curative strategies. Current medications for BPD are long-term, depending on the severity of the disorder and the patient’s response to the treatment. However, emerging drug therapy approaches for BPD are aiming to achieve longer periods of remission [82]. The risk genes and variants identified in this study are useful in leveraging pharmacogenomic research for drug design optimization that will target the cause of BPD at the molecular level, which is contrary to most drugs that only deal with the cellular ramifications of the disorder. Targeting the location of the genes dysregulated in BPD may provide better and longer curative effects because the root cause is directed and not the translated proteins themselves. The CRISPR technology is also one promising technique for gene-editing and -regulation, and its advancement serves as a breakthrough in the field of pharmacogenomics [83]. The risk genes and variants found in this study may serve as a preliminary step in determining the guide RNA sequences that could be used for editing the genome of patients with BPD. Additionally, gene replacement, gene silencing, or gene knock-out may be employed to edit these BPD-causing genes and variants [84]. However, research on gene therapy is still ongoing, but despite the current limitations of this method, its purpose still holds great promise in the future of pharmacogenomic techniques for engineering curative strategies for BPD.

Using a computational and pharmacological assessment of the transcriptome of BPD, we have found that the nAcc region is involved in the expression of genes responsible for the deregulated transcription of neurotransmitters, while the DLPFC region is involved in the expression of genes responsible for neurotransmission impairment. These results primarily underscore the neurobiological aspect of BPD, which could further enlighten the molecular and cellular factors that lead to the phenotypic manifestations of the disorder. Aside from that, among the risk genes identified, *DRD2* was determined to be greatly associated with the nAcc, *GRIK4* with the AnCg, and *IGFBP2* with the DLPFC, and most of them showed the presence of SNPs.

## 5. Conclusions

The complex nature of BPD is one reason why the study of its etiology is challenging, just like other psychiatric conditions. However, advances in laboratory technology, such as NGS, paved the way for researchers to better understand the transcriptome of this disorder, enlightening the biological mechanisms that drive the severity of its manifestations. In this study, we performed intramodular connectivity analysis and network pharmacology assessment of the expression profiles of patients diagnosed with BPD to elucidate the molecular and cellular causes of the disorder. The results suggest that the nAcc was highly associated with the expression of genes responsible for the deregulated transcription of neurotransmitters. Meanwhile, the DLPFC was greatly correlated to genes involved in the impairment of components in neurotransmission. Relative to the two brain regions, the AnCg was not found to be highly correlated to any of the identified expressions, though it showed considerable association with other mechanisms evident in BPD. Disease-associated variants were also found present among the significant genes of the disorder and may be targeted for innovating better treatment strategies.

Future objectives may investigate other brain regions that would better elucidate the neurobiological manifestations of BPD. The SNPs reported in this study may also be further evaluated to expound on the genetic connectivity of BPD with other psychiatric disorders. This study only covered the genetic factors affecting BPD; thus, it would greatly benefit the current literature if the environmental and epigenetic conditions that influence the disorder were tackled as well. Finally, DEA may be employed to identify which genes may be upregulated or downregulated, which could provide more insights into future transcriptomic studies of BPD. We also emphasize that post-mortem brain samples were used in this study, and that may incur biological limitations posed by certain inevitable factors such as PMI, patient history, or ethnicity. The data were also initially analyzed through in vitro methods and then assessed using in silico computations, so the pharmacological findings in this study may require further in vivo experimentations or be subjected to advanced clinical research. In the future, as more transcriptomic data become available, the complex interactions among genes, RNAs, and proteins will become clearer and eventually lead to a better understanding of life.

## Figures and Tables

**Figure 1 biology-13-00787-f001:**
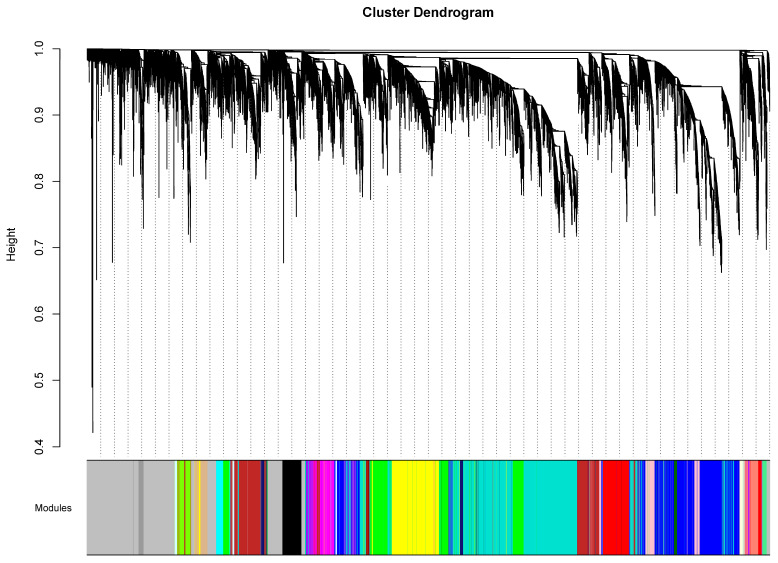
Dendrogram based on the hierarchal clustering of genes from which the modules that display high gene interconnectivity were identified.

**Figure 2 biology-13-00787-f002:**
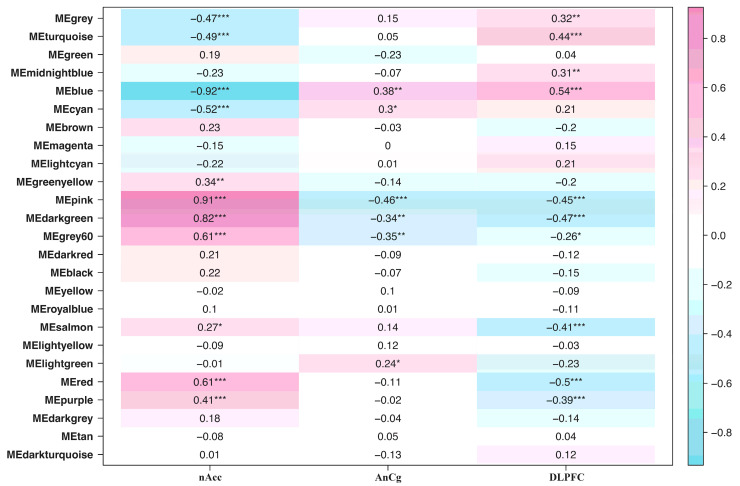
Network heatmap of the identified modules. Darker colors indicate greater connectivity; positive correlation means strong association among the genes in that module; otherwise, negative correlation suggests weak association. Moreover, the number of asterisks indicates the likelihood that the expression patterns identified in that module greatly characterize that brain region.

**Figure 3 biology-13-00787-f003:**
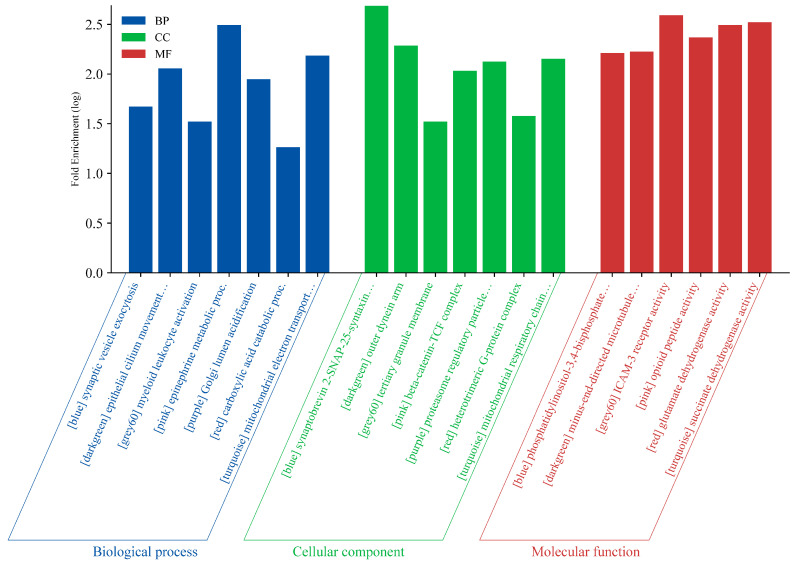
Bar plots of the top clusters from each annotation of the 7 modules identified to have significant expression among the three brain regions.

**Figure 4 biology-13-00787-f004:**
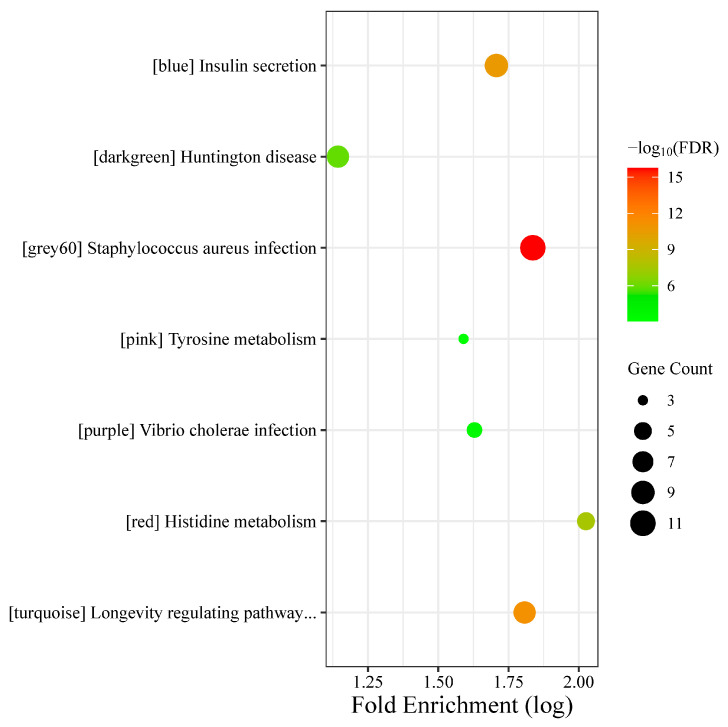
Bubble plot visualization of the top cluster in the KEGG pathway analysis based on their respective fold enrichment score.

**Figure 5 biology-13-00787-f005:**
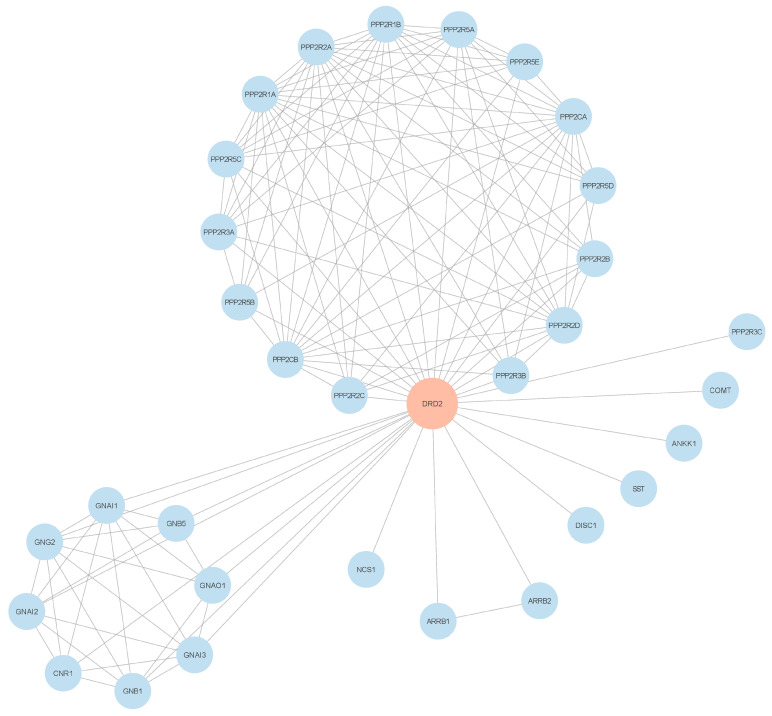
Modified PPI network of *DRD2* constructed from the expression profiles of BPD patients, showing the interactions of the gene with its first-degree neighbors.

**Table 1 biology-13-00787-t001:** Identified genes among the three brain regions which showed the presence of disease-associated variants.

Sample	Gene	Variant	Type
nAcc	*DRD2*	rs1801028	missense variant
	*GFRA2*	rs7833426	intron variant
	*DCBLD1*	rs62433108	intron variant
AnCg	*ST8SIA2*	rs4777989	intron variant
	*ADAMTS16*	rs16875288	intron variant
DLPFC	*FOXO3*	rs1536057	intron variant
		rs1935952	
		rs2802292	
	*ITGA9*	rs166508	intron variant
	*CUBN*	rs7904579	intron variant
	*PLCB4*	rs2299682	intron variant
	*RORB*	rs1327836	intron variant

## Data Availability

The original data presented in the study are openly available in GEO-NCBI at 10.1186/s13073-017-0458-5 or GSE80655.

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
