# Peer review of "Codes between Poles: Linking Transcriptomic Insights into the Neurobiology of Bipolar Disorder"

_biology, 2024, doi:10.3390/biology13100787_

Round 1

Reviewer 1 Report

Comments and Suggestions for Authors

The major problem of the manuscript is that this is just a bioinformatics analysis of the previously published data. 

Comments on the Quality of English Language

The grammar of English should be revised.

Reviewer 2 Report

Comments and Suggestions for Authors

The manuscript provides a comprehensive analysis of the molecular and cellular mechanisms underlying bipolar disorder (BPD) by analyzing gene expression from post-mortem brain tissues. The study focuses on three regions of the brain: the nucleus accumbens (nAcc) which was found to be highly correlated with genes involved in neurotransmitter deregulation; anterior cingulate cortex (AnCg) which showed weaker correlations with gene expressions compared to nAcc and DLPFC; and dorsolateral prefrontal cortex (DLPFC) which was strongly associated with genes crucial for neurotransmission, highlighting its role in BPD pathology. Additionally, several BPD-associated genetic variants were identified, suggesting that BPD shares genetic links with other psychiatric conditions. Overall, the study enhances the understanding of the neurobiology of bipolar disorder through transcriptomic analysis and provides a foundation for future research on targeted drug development. To gain deeper insights into the molecular mechanisms of bipolar disorder and provide stronger foundations for therapeutic development and precision medicine approaches, the following analysis could be included:

1. The demographic details of the post-mortem samples are not mentioned in detail. Performing differential gene expression analysis stratified by BPD subtypes (e.g., Bipolar I, Bipolar II) could reveal subtype-specific molecular signatures, providing insights into the heterogeneity of BPD.

2. The use of post-mortem brain tissue introduces limitations related to tissue preservation and the potential for post-mortem changes that might not accurately reflect living brain dynamics. The manuscript could discuss potential biases introduced by this approach or complement the findings with data from living subjects.

3. Besides genetic factors that contribute to BPD, the study could benefit from a deeper exploration of environmental and epigenetic contributions to BPD, providing a more holistic view of how both genetic and non-genetic factors influence the disorder.

4. While the manuscript discusses potential therapeutic targets, it does not provide specific details on how these findings could be directly translated into clinical treatments. The manuscript could provide more detailed discussions on how the identified genetic variants and pathways could be leveraged for drug development and treatment strategies.

5. The study could improve by offering a more detailed mechanistic explanation of how the identified genes and pathways contribute to BPD's pathophysiology.

Minor:

1. Line 40, “BFD”, should be “BPD”

2. The resolution of Figure 5 are too low to read.

Comments on the Quality of English Language

Minor editing of English language is required.

Reviewer 3 Report

Comments and Suggestions for Authors

The manuscript entitled “Codes Between Poles: Linking Transcriptomic Insights into the Neurobiology of Bipolar Disorder” is devoted to determine biological pathways responsible for the manifestation of bipolar disorder based on previous RNA-seq data from 24 patients obtained from the NCBI-GEO database. The manuscript is a well-structured and a well-written paper, which implies a specific approach such as intramodular connectivity analysis to unravel the differences in the gene expression of post-mortem bipolar disorder samples obtained from three brain regions related to this disease. The study represents a relevant study, which can help to improve the existing studies in this field via suggesting a new research pipeline. In addition, the manuscript is characterized by a detailed description of bioinformatics procedures and the functions, which were used to perform all the analyses in R, as well as a comprehensive number of tables and figures.

However, several issues need to be clarified.

1.     I suggest to add more details on the results revealed in the study in the Abstract, i.e. top differentially expressed genes or mention the genes previously linked to the disease as stated in Table 1.

2.     It would be more appropriate to use in the Table S2 the same designations of brain regions as they are stated in the whole text (i.e. nAcc, etc.) or explain abbreviations NA, ACC, DPC in the Notes to the table.

3.     I suggest to add one paragraph in the Discussion, which briefly summarizes the main findings revealed in the present study, and one paragraph in the Conclusion section to mention the possible limitations of the study.

4.     The quality of some figures is rather insufficient to read the aliases of genes obtained from the Cytoscape.

5.     The following lines should be checked for correctness:

-line 40: Correct BFD into BPD;

- lines 469, 474: “an SNP”;

- line 376: “highly cognitive functions” have to be changed into “higher cognitive functions”.

I would suggest to accept after minor revision, providing that the authors addressed all the comments.

Comments on the Quality of English Language

Minor English checking is required.
